# The Risk and Clinical Implications of Antibiotic-Associated Acute Kidney Injury: A Review of the Clinical Data for Agents with Signals from the Food and Drug Administration’s Adverse Event Reporting System (FAERS) Database

**DOI:** 10.3390/antibiotics11101367

**Published:** 2022-10-06

**Authors:** Kalin M. Clifford, Ashley R. Selby, Kelly R. Reveles, Chengwen Teng, Ronald G. Hall, Jamie McCarrell, Carlos A. Alvarez

**Affiliations:** 1Department of Pharmacy Practice, Jerry H. Hodge School of Pharmacy, Texas Tech University Health Sciences Center, Dallas, TX 75235, USA; 2College of Pharmacy, The University of Texas at Austin, Austin, TX 78701, USA; 3Pharmacotherapy Education and Research Center, University of Texas Health San Antonio, San Antonio, TX 78229, USA; 4Department of Clinical Pharmacy and Outcomes Sciences, College of Pharmacy, University of South Carolina, Columbia, SC 29208, USA; 5Department of Pharmacy, Baptist St. Anthony’s Health System, Amarillo, TX 79106, USA

**Keywords:** acute kidney injury, adverse drug events, antibiotics, nephrotoxicity

## Abstract

Antibiotic-associated acute kidney injury (AA-AKI) is quite common, especially among hospitalized patients; however, little is known about risk factors or mechanisms of why AA-AKI occurs. In this review, the authors have reviewed all available literature prior to 1 June 2022, with a large number of AKI reports. Information regarding risk factors of AA-AKI, mechanisms behind AA-AKI, and treatment/management principles to decrease AA-AKI risk were collected and reviewed. Patients treated in the inpatient setting are at increased risk of AA-AKI due to common risk factors: hypovolemia, concomitant use of other nephrotoxic medications, and exacerbation of comorbid conditions. Clinicians should attempt to correct risk factors for AA-AKI, choose antibiotic therapies with decreased association of AA-AKI to protect their high-risk patients, and narrow, when clinically possible, the use of antibiotics which have decreased incidence of AKI. To treat AKI, it is still recommended to discontinue all offending nephrotoxic agents and to renally adjust all medications according to package insert recommendations to decrease patient harm.

## 1. Introduction

Acute kidney injury (AKI) is defined as the sudden decrease in kidney function over hours to days, with or without kidney damage. AKI is common among patients in all health care settings, including approximately 20% of hospitalized patients and 40% of patients in intensive care units (ICUs) [1,2]. The majority of antibiotics commonly used in clinical practice can contribute to AKI (See Figure 1). The incidence of drug-induced AKI may be as high as 60%, and antibiotics are considered a main contributor [3,4,5].

AKI can also impact other important clinical outcomes. Of hospitalized patients who develop AKI, 10% require renal replacement therapy even after discontinuing the causative medications [1]. It has been shown to result in a 50% mortality rate for severe cases requiring renal replacement therapy [1]. There is also growing evidence that quicker identification of AKI does not improve time to resolution of AKI, and it can permanently change baseline serum creatinine (SCr) levels [5,6]. This information magnifies the importance of the prevention of AKI given that we currently lack effective AKI treatment strategies. 

Antibiotics have been called the “fire extinguishers” of medicine and are often necessary in spite of the potential harms associated with AKI, even in patients who have AKI risk factors. This comprehensive review describes the association between AKI and antibiotics or antibiotic classes, including the mechanism or action, common uses, clinical AKI data, potential mechanisms of causing AKI, other risk factors for AKI, and summary recommendations. 

## 2. An Overview of Acute Kidney Injury

The three most commonly used classifications for diagnosing AKI in the medical literature are the (1) Risk, Injury, Failure, Loss of Kidney Function, End-Stage Kidney Disease Classification (RIFLE), (2) Acute Kidney Injury Network Classification (AKIN), and the (3) Kidney Disease Improving Global Outcomes (KDIGO) Classification. The different breakpoints for SCr or glomerular filtration rate (GFR), and urine output (UO) when classifying AKI severity are shown in Table 1. RIFLE was the first generally accepted attempt to achieve a consensus definition of AKI rather than provider-specific diagnostic criteria [7]. The AKIN classification was created due to additional, cumulative evidence that more minute increases in SCr were also associated with poor outcomes, and variation between multiple hospitals on the initiation of renal replacement therapy and dialysis [8,9,10,11]. The AKIN classification improved the specificity and sensitivity of AKI diagnosis, but did not improve AKI diagnostic acuity [12]. KDIGO merged RIFLE and AKIN to provide criteria to be used in daily clinical practice and research. Gameiro and colleagues found that KDIGO increased the reported AKI incidence and outcome prediction compared to AKIN and RIFLE [13].

The most common risk factors for AKI are shown in Table 2. Antibiotics, as a drug class, have often been associated with AKI; however, each antibiotic and/or class have different risk profiles. A previous evaluation of the Food and Drug Administration (FDA) Adverse Event Reporting System (FAERS) identified 14 different antibiotic classes or specific antibiotics associated with AKI, which will serve as the guide for this review [14].

## 3. Specific Antibiotics Classes and Agents

Search strategy. For this narrative review, the authors utilized the PubMed database to search for antibiotics and antibiotic classes that were specifically associated with acute kidney injury in the FAERS database analysis by Patek and colleagues, acute kidney injury, kidney failure, and renal failure [14]. The FAERS database is a United States government-funded dashboard that provides results of all reported adverse events due to medications, biologics, medical devices, food, and other products regulated by the FDA. For adverse events caused by medications, the information is pulled from MedWatch (FDA Safety Information and Adverse Event Reporting Program), but this information can be completed not just by health care providers, but also manufacturers, distributors, and anyone (including patients and caregivers) who experience, or witness, an adverse drug event. It is a useful tool for epidemiological and pharmacovigilance studies, as manufacturers are required to report any adverse event caused by their product within 14 days. This data should also be viewed with caution, as many reports can be incomplete due to missing clinical information that may not be known, or understood, by the general public. Other limitations of FAERS include no ability to prove causal relationship between the reported exposure and the medication, potential for biases from healthcare providers, and the fact that not all adverse events are reported, as it voluntary for healthcare providers and the general population to complete these reports. Nevertheless, this tool can identify adverse reactions and medications for further research to be performed. The authors chose to include information from randomized controlled trials, systematic reviews, and larger case series. Nephrotoxicity rates from case reports and smaller case series were excluded; however, information regarding mechanism of nephrotoxicity was included. After the literature review, the authors found the article of Patek and colleagues which used FAERS to identify antibiotic-associated nephrotoxicity rates. The authors use the Patek results to organize the antibiotics, and classes, based on their likelihood of causing AKI, reported as relative odds ratios with 95% confidence intervals. Table 3 provides this information as well as additional incidence data and potential causes of AKI.

### 3.1. Polymyxins (FAERS Relative Odds Ratio (ROR) 33.10, 95% Confidence Interval (CI) 21.24–51.56)

#### 3.1.1. Mechanism of Action

Polymyxins act primarily by binding to membrane phospholipids and disrupting the bacterial cytoplasmic membrane.

#### 3.1.2. Common Uses

The FDA-approved indication for both colistin and polymyxin B are the treatment of acute or chronic infections caused by sensitive strains of some Gram-negative bacilli (specifically *Pseudomonas aeruginosa*) that are resistant to other antibiotics or in patients who are allergic to other antibiotics. Polymyxins are also commonly considered a ‘last-line’ therapy for infections caused by multi-drug resistant (MDR) Gram-negative pathogens.

#### 3.1.3. Clinical Data Regarding AKI

Polymyxins are major contributors to antibiotic-associated AKI and polymyxin B has a FDA Black Box Warning for AKI. The earliest meta-analysis (2010–2015) found no significant association between colistin use and AKI, but more recent meta-analyses in 2018 (odds ratio [OR] 2.50, 95% CI 1.05–5.98], 2020 (OR 1.82, 95% CI 1.13–2.92), and 2021 (OR 2.23, 95% CI 1.58–3.15) demonstrated a significant association [14,15,16]. Six clinical studies have compared AKI rates, and five have observed worse AKI rates/severity with colistin compared to polymyxin B.

For those patients requiring colistin or polymyxin B, investigators discovered that AKI occurred more often with once-daily dosing compared to twice-daily dosing (47% versus 17%, *p* = 0.0005) [17]. This may indicate that more frequent dosing decreases AKI risk, but the results of this study differ from those found in previous animal models. Higher quality data are needed to identify the optimal dosing strategy to reduce AKI other adverse effects. 

#### 3.1.4. Proposed Mechanism of AKI

While not completely elucidated, the primary mechanism of AKI associated with polymyxins is likely due to damage to the cell membrane of proximal tubule cells due to colistin’s detergent activity. A secondary mechanism may be due to the accumulation of colistin causing damage to mitochondria within proximal tubule cells [18]. 

#### 3.1.5. Other Potential Risk Factors

The polymyxin dose used is the primary medication-related factor. Patient factors include age, chronic kidney disease, elevated baseline SCr level, and chronic liver disease. The use of concomitant loop diuretics (i.e., furosemide, bumetanide, torsemide) and/or vancomycin can also increase AKI risk [19,20,21,22,23].

#### 3.1.6. Summary

The AKI incidence for polymyxins is 8–60%. The use of polymyxins should be avoided whenever possible, as newer agents have been shown to be more protective against AKI and produce better clinical cure rates [24]. There are other agents in most situations that should be used instead of polymyxins. The consensus guidelines provide dosing recommendations for patients who require this last-line therapy.

### 3.2. Aminoglycosides (FAERS ROR 17.41, 95%CI 14.49–20.90)

#### 3.2.1. Mechanism of Action

Aminoglycosides inhibit protein synthesis by binding, with high affinity, to the A-site on the 16S ribosomal ribonucleic acid of the 30S ribosome.

#### 3.2.2. Common Uses

Aminoglycosides are used in treating a variety of infections including bacteremia, hospital-acquired pneumonia (HAP), and nontuberculous mycobacterial infections. The primary use of aminoglycosides in recent years has been in cases of MDR pathogens. The clinical utility of aminoglycosides is limited to combination therapy, with the exception of urinary tract infections.

#### 3.2.3. Clinical Data Regarding AKI

The 2012 KDIGO have stated to avoid aminoglycosides in patients at risk of AKI due to their nephrotoxic effects [25,26,27,28]. The risk of AKI with aminoglycosides has been reported to be in the range of 5–25% [25,27,29,30,31,32,33]. Once a patient has AKI due to an aminoglycoside, their renal function may not recover to baseline [25,29,34]. 

#### 3.2.4. Proposed mechanism of AKI

The mechanism behind their nephrotoxicity includes accumulation in the proximal tubular cells causing apoptosis and acute tubular necrosis (ATN), due to disrupting phospholipid metabolism [25,26,27,35].

#### 3.2.5. Other Potential Risk Factors

Oliveira and colleagues performed univariate analyses on various risk factors on AKI occurrence [36]. Logistic regression analyses noted risk factors including diabetes (OR 2.13, 95% CI 1.01–4.49), the use of iodinated contrast (OR 2.13; 95% CI 1.02–4.43), hypotension (OR 1.83, 95% CI 1.14–2.94), and the concomitant use of other nephrotoxic agents (OR 1.61; 95% CI 1.00–2.59) [36]. Other risk factors for aminoglycoside-associated AKI include older age, chronic kidney disease, liver disease, hypotension, hypoalbuminemia, duration of therapy, high trough and peak levels, cumulative dose and multiple doses administered daily [25,27,28,29,32,33,35,36,37,38]. 

#### 3.2.6. Summary

Aminoglycosides are one of the most commonly recognized AKI-inducing agents within clinical practice today. Clinicians should conduct a thorough risk-benefit assessment in patients with risk factors to ensure their use is necessary. Frequent monitoring and use of extended interval dosing is needed to help minimize the risk of AKI in patients who require these agents.

### 3.3. Vancomycin (FAERS ROR 15.28, 95%CI 13.82–16.90)

#### 3.3.1. Mechanism of Action

Vancomycin inhibits transpeptidation by binding to D-alanyl-D-alanine residues of the bacterial cell wall.

#### 3.3.2. Common Uses

Vancomycin is commonly used as a component of empiric antimicrobial therapy for hospitalized patients suspected of having methicillin-resistant Staphylococcus aureus (MRSA) infections and as definitive therapy for confirmed MRSA infections. It can also be used for other Gram-positive pathogens, particularly those resistant to other antimicrobials. 

#### 3.3.3. Clinical Data Regarding AKI

##### Therapeutic Drug Monitoring

Guidelines for many infectious disease states recommend a therapeutic range between 15–20 micrograms/milliliter (mcg/mL) [39,40,41,42,43,44,45,46]. Trough concentrations greater than 15 milligrams/liter (mg/L) have been associated with AKI [47]. A 5 mcg/mL increase in vancomycin trough concentration is associated with a higher risk of AKI (OR 1.42; 95% CI 1.10–1.82) [48]. The new vancomycin consensus guidelines recommend area under the curve (AUC)-based monitoring with a goal of 400–600 [49]. This recommendation is largely based on a decreased AKI rate compared to goal troughs of 15–20 in pragmatic clinical trials and retrospective data. Others have recently suggested that a goal trough <15 mg/L may achieve similar safety and effectiveness of AUC-guided dosing without the additional cost, but direct comparison of these two approaches are lacking [50].

##### Additional Dosing Strategies

Using four grams or more of vancomycin has also been linked with AKI. This approach was only evaluated in a small number of patients who received a high milligram/kilogram (mg/kg) dose. More frequent dosing has not consistently shown an increased risk in AKI [51,52,53]. The use of loading doses for osteomyelitis, pneumonia, or endocarditis has not increased AKI rates [54,55,56]. The use of continuous infusion for administration may decrease AKI, while others have shown similar risk with traditional infusions [57,58,59,60]. 

##### Impact of Duration of Therapy

The overall risk of AKI appears to increase after four days of therapy and continues to increase as treatment duration increases [41,42,43,52,61,62,63,64]. Specifically, when the duration of therapy was extended from seven and fewer days to greater than 8 days, the rate of AKI occurrence increased from 6% to 21% [40]. Many of these studies may have the limitation of observation bias, specifically that patients receiving vancomycin for longer durations were evaluated for AKI for a longer period of time.

##### Relative AKI Risk to Other Anti-MRSA Agents

Vancomycin was associated with a higher risk of AKI (OR 2.45, 95% CI 1.69–3.55) compared to other antibiotics in a meta-analysis [65]. Studies have demonstrated a lower AKI risk with other MRSA antibiotics (e.g., linezolid, daptomycin) compared to vancomycin (OR 0.31, 95% CI 0.13–0.74) [66]. When reviewing patients with nosocomial MRSA pneumonias, a lower AKI risk was seen with linezolid compared to vancomycin (RR 0.50, 95% CI 0.31–0.8) [67]. A randomized controlled clinical trial found that vancomycin was significantly associated with AKI compared to linezolid (18.2 vs. 8.4%) [68].

The use of continuous-infusion dosing, AUC:minimum inhibitory concentration (MIC)-based dosing strategies, and increased therapeutic drug monitoring are all potential methods to decrease vancomycin-associated AKI [57,58,69,70,71,72]. The Infectious Disease Society of America has approved a new dosing guideline for vancomycin which includes many of these strategies [49]. Many hospitals and other healthcare settings are still attempting to determine if the logistical and educational challenges associated with implementing these recommendations are worthwhile [73]. 

There have been studies evaluating whether alternative MRSA agents should be used over vancomycin in patients with additional risk factors for AKI [74]. This concept has not been studied with clinical cure as its outcome, only when evaluating for AKI incidence [75].

#### 3.3.4. Proposed Mechanism of AKI

The mechanism by which AKI occurs is not fully understood and multiple pathways have been postulated. Vancomycin has been known as “Mississippi Mud” due to the impurities noted in early formulations, which were originally thought to be the main causes of its nephrotoxic effect. By the 1980s, advances in manufacturing helped improve its purity and decreased its AKI rates [76]. Animal models suggested that vancomycin causes proinflammatory oxidation and free radical formation within the proximal tubule of the kidneys, which then contributes to mitochondrial dysfunction and cellular apoptosis [40,47,59,77,78,79]. Additional findings have discussed the formulation of vancomycin tubular casts, with concurrent acute tubular necrosis (ATN) and acute interstitial nephritis (AIN) [80,81,82,83].

#### 3.3.5. Other Potential Risk Factors

Most studies evaluating vancomycin-associated AKI found that intensive care unit residence, hypotension requiring the use of vasopressors, elevated acute physiology and chronic health evaluation (APACHE) II scores, Pitt bacteremia scores of 4 or greater, and increased Charlson comorbidity indices are associated with AKI [39,41,43,45,64]. Other risk factors for AKI include baseline renal dysfunction, obesity, and older age. Jorgensen and colleagues have identified chronic alcohol abuse and lack of medical insurance as other independent risk factors of vancomycin-associated nephrotoxicity [84].

The use of concomitant nephrotoxic agents with vancomycin has increased further incidence of AKI. Amphotericin B, aminoglycosides, and tacrolimus are the most commonly associated [40,59,85,86]. Angiotensin-converting enzyme inhibitors (e.g., lisinopril, enalapril) and loop diuretics also have been associated with AKI due to their direct effects within the nephron and contribution to overdiuresis [87,88]. Other medications that may increase AKI risk include tenofovir and foscarnet [89,90]. 

#### 3.3.6. Summary

Vancomycin is often considered the second most commonly recognized AKI-inducing agent within clinical practice today. Vancomycin-induced nephrotoxicity is also one of the most studied adverse reactions for which there is robust evidence to be reviewed. Despite the robust evidence, most randomized controlled trial data with vancomycin did not utilize standardized AKI definitions. Clinicians should conduct a thorough risk–benefit assessment in patients with risk factors to ensure their use is necessary. The utility of therapeutic drug monitoring and use of alternative dosing strategies (e.g., continuous infusion) are still debated by experts in the field. Many also suggest vancomycin avoidance as a viable AKI prevention strategy.

### 3.4. Trimethoprim/Sulfamethoxazole (FAERS ROR 13.72, 95%CI 11.94–15.76)

#### 3.4.1. Mechanism of Action

Trimethoprim/sulfamethoxazole (TMP/SMX) is a bactericidal combination antibiotic that acts by inhibiting the bacterial folic acid synthesis pathway. 

#### 3.4.2. Common Uses

TMP/SMX is most commonly used for urinary tract infections and skin and soft tissue infection, especially since the emergence of community-acquired MRSA. 

#### 3.4.3. Clinical Data Regarding AKI

TMP/SMX is generally well-tolerated. Several studies and case reports have identified changes in renal function, nephrotoxicity, and AKI, especially when using increased TMP/SMX to eradicate more aggressive infections. 

It was found that when comparing the use of trimethoprim, ciprofloxacin, and amoxicillin in an elderly population, the highest risk of AKI was amongst older adults who received trimethoprim compared to amoxicillin (OR 1.72; 95% CI 1.31–2.24) [91]. The authors noted low absolute risk for AKI—for every 1000 urinary tract infections treated, use of TMP would result in two additional AKIs compared to amoxicillin [91]. 

TMP/SMX-associated AKI appears to be a dose-dependent phenomenon. High-dose TMP/SMX (at least four double-strength tablets per day) was associated with AKI (3.67% vs. 1.63%, *p* = 0.044) when compared to standard dose TMP/SMX therapy (OR 3.70; 95% CI 2.06–5.14) [92]. This increased AKI risk was also seen by Gentry and colleagues in a retrospective cohort study which also found that those who received a high dose of TMP/SMX (2.0% vs. 0.7%, *p* = 0.0001) compared to the standard dose [92].

#### 3.4.4. Proposed Mechanism of AKI

The mechanism by which AKI occurs is multifaceted. The four mechanisms that have been postulated at this time are AIN, ATN, crystalluria forming within the distal convoluted tubule, and a reversible inhibition of the tubular secretion of creatinine which causes elevated SCr levels without affecting GFR [93,94,95,96,97]. 

#### 3.4.5. Other Potential Risk Factors

Other risk factors for AKI in patients receiving TMP/SMX include concomitant angiotensin converting enzyme inhibitor (OR 2.36, 95% CI 1.01–5.24), potassium supplementation (OR 4.10, 95% CI 1.45–10.10), and elevated baseline SCr (OR 2110, 95% CI 724–7980) [92]. Diabetes mellitus and hypertension were independent predictors of AKI in a veteran population [97].

#### 3.4.6. Summary

TMP/SMX does not meaningfully increase the risk of AKI in most patients as the increase in risk is very low. Clinicians should use caution when prescribing TMP/SMX in those with AKI risk factors including pre-existing conditions, renal disease, and other medications associated with AKI.

### 3.5. Penicillin Combinations (FAERS ROR 7.95, 95%CI 7.09–8.91)

#### 3.5.1. Mechanism of Action

Penicillin antibiotics kill bacteria by inhibiting the synthesis of bacterial cell walls. They bind preferentially to specific penicillin-binding proteins located inside bacterial cell walls. Beta-lactamase inhibitors are given in combination with penicillins to prevent inactivation of the beta-lactam ring by serine beta-lactamases produced by some Gram-negative bacteria. 

#### 3.5.2. Common Uses

Penicillin combinations are commonly used for diabetic foot infections, HAP, and intra-abdominal infections (IAI).

#### 3.5.3. Clinical Data Regarding AKI

A very small retrospective study observed AKI rates in the range 1.7–38.5%, with higher rates for 4.5 g doses compared with 2.25 g doses [98,99,100]. A study evaluating piperacillin-tazobactam as monotherapy for patients with Gram-negative bacteremia reported an AKI rate of 13.1%. The same study suggested that the duration of therapy for piperacillin-tazobactam did not significantly impact nephrotoxicity [101]. Prolonged infusion of piperacillin-tazobactam has also not been associated with AKI (22 vs. 19%, *p* = 0.1) [102].

Rutter and colleagues found that the data did not support an increased risk of AKI for ampicillin-sulbactam when compared with piperacillin-tazobactam (9 vs. 11%, *p* = 0.14) [103]. Inadequate amoxicillin administration was noted as a risk factor for crystal nephropathy in a pharmacovigilance study [104].

#### 3.5.4. Proposed Mechanism of AKI

Hypersensitivity reaction leading to interstitial nephritis and crystalluria with tubular injury are considered the two primary mechanisms for the development of AKI [105,106,107,108,109]. More recent case reports describe acute tubulointerstitial nephritis with or without hemorrhagic cystitis [110]. 

#### 3.5.5. Other Potential Risk Factors

Risk factors for amoxicillin-induced crystal nephropathy include older age and endocarditis. While using the increased dose of piperacillin-tazobactam, concomitant use of gentamicin with penicillin and its derivative both increase the risk of penicillin-associated nephrotoxicity.

#### 3.5.6. Summary

Penicillin combinations are generally considered safe as monotherapy and have modest rates of AKI. The duration of therapy or type of beta-lactamase inhibitor has not been associated with AKI.

### 3.6. Clindamycin (FAERS ROR 6.46, 95%CI 5.18–8.04)

#### 3.6.1. Mechanism of Action

Clindamycin’s main mechanism of action is to inhibit bacterial protein synthesis by binding at the 50S ribosomal subunit. 

#### 3.6.2. Common Uses

Clindamycin is commonly used for skin and soft tissue infections involving Gram-positive pathogens and for suppression of toxin production. Its use increased with the increased incidence of community-acquired MRSA.

#### 3.6.3. Clinical Data Regarding AKI

Two retrospective analyses from China, in addition to the FAERS analysis, within our review, reported 24 cases in one analysis and 50 cases in the second [5]. In the first analysis, the cases were severe, requiring renal replacement therapy (75% required continuous renal replacement therapy, 12.5% required intermittent dialysis) [5]. However, in both analyses, renal function significantly recovered (SCr < 1.24 mg/dL) within two months of discharge [5,6].

#### 3.6.4. Proposed Mechanism of AKI

While there are reports of AKI with clindamycin [15], the mechanism has not been established. Retrospective studies have suggested an association with biopsy-proven AIN and ATN as well as crystal nephropathy leading to tubular toxicity [67,111]. 

#### 3.6.5. Other Potential Risk Factors

Potential risk factors include gross hematuria, anemia, and hypertension [5]. One AKI episode within a case series documented the clinical presentation of their patient was quite similar to main adverse effects of clindamycin, specifically nausea, vomiting, and abdominal discomfort [5,112]. 

#### 3.6.6. Summary

The true AKI incidence for clindamycin is unknown, which likely means the risk of AKI is rare [112]. The lack of a systematic approach to evaluating AKI risk does not account for the risk of diarrhea and resulting hypovolemia that increase the risk of AKI.

### 3.7. Cephalosporins (FAERS ROR 6.07, 95%CI 5.23–7.05)

#### 3.7.1. Mechanism of Action

Cephalosporins act through inhibition of bacterial cell wall synthesis by binding at least one penicillin-binding protein. 

#### 3.7.2. Common Uses

The cephalosporin antibiotic class consists of five generations and nearly 20 bactericidal agents. Each generation has a unique spectrum of activity. First and second generation cephalosporins are used to treat acute bacterial skin and skin-structure infections caused by Gram-positive pathogens such as methicillin-sensitive Staphylococcus aureus and Streptococcus spp. Later generations have more activity against Gram-negative pathogens and are commonly prescribed for bacteremia, IAI, community-acquired pneumonia (CAP), and HAP.

#### 3.7.3. Clinical Data Regarding AKI

Renal adverse effects of cephalosporins have been documented for five decades since cephaloridine, the first cephalosporin antibiotic was approved for use in the United States, but now is no longer on the market today [113]. Although more recent additions to the medication class have fewer reported renal adverse effects, several case reports indicate a multifaceted risk still exists.

#### 3.7.4. Proposed Mechanism of AKI

Proposed mechanisms for cephalosporins contributing to AKI include ATN and AIN [114,115,116,117,118,119]. Some case reports have also identified immune hemolytic anemia within the setting of cephalosporin-associated AKI. Cases of postrenal ureteric calculi have been reported with the use of ceftriaxone, but the pathophysiology is not understood at this time [120,121].

#### 3.7.5. Other Potential Risk Factors

Surgery is a risk factor for AKI. Concomitant aminoglycoside use may increase AKI risk; however, Pannell and colleagues reported no difference in AKI in 159 patients with open fractures given cefazolin with or without gentamicin (4% versus 4.8%, *p* = 0.599) [122]. Tucker and colleagues similarly evaluated cefuroxime with or without gentamicin in 2560 patients receiving hip and knee arthroplasty (1.36% vs. 1.07%, *p* = 0.524) [123].

Ceftolozane/tazobactam, a combination cephalosporin with beta-lactamase inhibitor has a much lower AKI rate (3.8%) than polymyxin or aminoglycosides (adjusted OR 0.08, 95% CI 0.03–0.22) [124]. The incidence of AKI was not impacted by length of infusion. This is similar to Cotner and colleagues who found no difference in AKI rates for intermittent vs. prolonged-infusions for beta-lactams in univariable (21.6% versus 18.6%, *p* = 0.1) or multivariable analyses (OR 1.07, 95% CI 0.83–1.39) [102]. 

#### 3.7.6. Summary

There is a low risk of AKI occurrence with cephalosporins. These agents are commonly used in place of other antimicrobials with a higher risk of AKI (e.g., polymyxins, aminoglycosides).

### 3.8. Daptomycin (FAERS ROR 6.07, 95%CI 4.61–7.99)

#### 3.8.1. Mechanism of Action

Daptomycin’s bactericidal effect is achieved by permeabilization and depolarization of the bacterial cell membrane.

#### 3.8.2. Common Uses

Daptomycin is used for bloodstream infections, including bacteremia and endocarditis, as well as skin and soft tissue infections. 

#### 3.8.3. Clinical Data Regarding AKI

Daptomycin has not been commonly associated with AKI, but has only been compared to linezolid and vancomycin. The incidence of AKI with daptomycin was 5–20%, while vancomycin was 25–65% summarized between multiple studies [125,126,127,128].

#### 3.8.4. Proposed Mechanism of AKI

There is not a known mechanism identified in clinical literature. 

#### 3.8.5. Other Potential Risk Factors

Other common risk factors included concomitant nephrotoxic agents, recent cardiothoracic surgery, intensive care treatment, older age, and number of comorbidities [129]. 

#### 3.8.6. Summary

Daptomycin is used for bloodstream infections and endocarditis which have a non-zero baseline risk of AKI. However, daptomycin does not appear to be associated with nephrotoxicity based on the currently available data, given that its risk of AKI is lower than other standard therapies it has been compared to previously.

### 3.9. Macrolides (FAERS ROR 3.60, 95%CI 3.04–4.26)

#### 3.9.1. Mechanism of Action

Macrolides bind to the 50S ribosomal subunit of susceptible microorganisms, thus interfering with microbial protein synthesis.

#### 3.9.2. Common Uses

The macrolide class of antibiotics includes azithromycin, clarithromycin, and erythromycin. Macrolides are commonly used to treat CAP, chronic obstructive pulmonary disease exacerbations, and otitis media infections including those caused by atypical bacterial pathogens. 

#### 3.9.3. Clinical Data Regarding AKI

The incidence of AKI with macrolides is quite rare. Gandhi and colleagues reported an AKI incidence of <0.5% in their retrospective analysis of almost 200,000 participants (clarithromycin 0.44% vs. azithromycin while taking a calcium channel blocker 0.22%) [130]. 

#### 3.9.4. Proposed Mechanism of AKI

All three agents have had biopsy-proven results of causing AIN in case reports [131,132,133]. Drug–drug interactions (e.g., statins, nifedipine, verapamil) are another mechanism for AKI in retrospective analyses [134]. The mechanism is that cytochrome P 450 3A4 enzyme inhibitors increase the hypotensive effect of the calcium channel blocker and further increase the risk of hypoperfusion and AKI [130]. A third potential mechanism is rhabdomyolysis contributing to AKI, which would be secondary to drug–drug interactions (e.g., statins or theophylline) [130,135].

#### 3.9.5. Other Potential Risk Factors

Risk factors include older adults, use of dihydropyridines (nifedipine, felodipine, amlodipine), chronic kidney disease, and use of cytochrome P 450 3A4 inhibitors [130]. 

#### 3.9.6. Summary

The incidence of AKI with macrolides is rare. Clinicians should be aware of the risk factors, with a special emphasis on the impact of drug–drug interactions, contributing to macrolide-associated AKI.

### 3.10. Linezolid (FAERS ROR 3.48, 95%CI 2.54–4.77) 

#### 3.10.1. Mechanism of Action

Linezolid inhibits bacterial reproduction by selectively binding to a site on the 23S ribosomal RNA of the 50S subunit, thereby preventing initiation complex formation with the 70S ribosomal subunit.

#### 3.10.2. Common Uses

Linezolid is used for skin and soft tissue infections, pneumonia, and various other infections due to Gram-positive pathogens. 

#### 3.10.3. Clinical Data Regarding AKI

The rate of AKIs in patients receiving linezolid is 6–10% compared to 18–35% for vancomycin [63,68,136]. A randomized, controlled trial that also found that vancomycin was significantly associated with AKI compared to linezolid (18 vs. 8%) [68].

#### 3.10.4. Proposed Mechanism of AKI

Linezolid is not commonly associated with AKI and a mechanism has not been discovered to date.

#### 3.10.5. Other potential Risk Factors

The risk factors identified were concomitant nephrotoxic agents, recent cardiothoracic surgery, requiring intensive care treatment, older age, and an increased number of comorbid conditions [129]. 

#### 3.10.6. Summary

Linezolid is not commonly associated with AKI. When AKI has been documented, it has occurred in patients with multiple risk factors (e.g., use of nephrotoxic agents, need for intensive care, and multiple comorbid conditions) at baseline for AKI. Its AKI incidence is less than vancomycin and similar to daptomycin.

### 3.11. Carbapenems (FAERS ROR 3.31, 95%CI 2.58–4.25)

#### 3.11.1. Mechanism of Action

Carbapenems exert their bactericidal activity through inhibition of cell wall synthesis by penetrating the cell wall of most Gram-positive and Gram-negative bacteria to reach penicillin-binding protein targets. 

#### 3.11.2. Common Uses

Carbapenems are most commonly used for HAP, diabetic foot infections, IAI and other infections due to MDR Gram-negative pathogens.

#### 3.11.3. Clinical Data Regarding AKI

Data using a standardized definition for AKI with carbapenems is limited. Imipenem–cilastatin–relebactam had lower rates of AKI per KDIGO (20.7% vs. 81.3%, *p* < 0.001) and AKI defined by RIFLE criteria compared with the combination of imipenem–cilastatin and colistin (0% vs. 25.0%) [137]. Doripenem had lower AKI rates compared to imipenem–cilastatin for ventilator-associated pneumonia (15.7% vs. 17.9%) [138]. Meropenem had a decreased risk of AKI compared to piperacillin–tazobactam in a retrospective study (25 vs. 14%, *p* = 0.00001) [102]. Meropenem trough concentrations >44.45 mg/L were associated with AKI in one study (*n* = 94), but this has not been confirmed [139]. No data were available regarding ertapenem and AKI incidence. 

#### 3.11.4. Proposed Mechanism of AKI

Imipenem is thought to cause oxidative stress leading to intracellular accumulation caused by renal organic anion transporters (OATs) that contribute to cytotoxicity and nephrotoxicity within the proximal tubule cells. However, OATs were inhibited by cilastatin, which decreased accumulation of imipenem and decreased nephrotoxicity. It is thought that the cilastatin component of imipenem–cilastatin may actually provide inhibition of OATs and decrease the risk of nephrotoxicity [140].

#### 3.11.5. Other Potential Risk Factors

No other risk factors have been identified within the clinical literature evaluated.

#### 3.11.6. Summary

Carbapenems have a low likelihood of being the primary cause of nephrotoxicity given the low number of reports and low rates reported in randomized, controlled trials prior to the use of standardized AKI definitions. Cilastatin may play a nephroprotective role, but higher quality studies are needed to confirm this approach.

### 3.12. Metronidazole (FAERS ROR 2.55, 95%CI 1.94–3.36)

#### 3.12.1. Mechanism of Action

Metronidazole works by passive diffusion into the cytoplasm of anaerobic bacteria where transport proteins such as ferredoxin transfer electrons to the nitro group of metronidazole forming a nitroso free radical. This creates a concentration gradient for intracellular transport of metronidazole where the free radical of metronidazole interacts with intracellular deoxyribonucleic acid (DNA), resulting in the inhibition of DNA synthesis and degradation and ultimately bacterial death.

#### 3.12.2. Common Uses

Metronidazole is most commonly used for IAIs and other infections requiring anaerobic coverage.

#### 3.12.3. Clinical Data Regarding AKI

One case report was published in 1994, but it lacked any evidence for diagnosis, mechanism, or risk factors [141].

#### 3.12.4. Proposed Mechanism of AKI

There is not a known mechanism identified in the clinical literature.

#### 3.12.5. Other Potential Risk Factors

There is not any clinical literature available to identify risk factors of AKI and metronidazole use.

#### 3.12.6. Summary

At this time, there are not any data linking metronidazole to nephrotoxicity in the medical literature.

### 3.13. Tetracyclines (FAERS ROR 1.73, 95%CI 1.26–2.36)

#### 3.13.1. Mechanism of Action

Tetracyclines are chiefly bacteriostatic. They exert their antimicrobial effect via protein synthesis inhibition.

#### 3.13.2. Common Uses

Tetracyclines are used for a wide variety of uses including acute bacterial skin and skin structure infections, CAP, acne, and tick-borne infections.

#### 3.13.3. Clinical Data Regarding AKI

Tetracyclines have a few cases reports documented where AIN did occur with tetracycline and doxycycline specifically. Minocycline in combination with colistin has resulted in lower AKI rates than colistin monotherapy (11.8% vs. 23.7%, *p* = 0.007) [51,142,143]. 

#### 3.13.4. Proposed Mechanism of AKI

AIN has been the primary mechanism documented in case reports. Minocycline demonstrates antioxidant properties, including the inhibition of caspase 1 and caspase 3 activation, inhibition of inducible nitric oxide synthase, and enhancement of Bcl-2-derived effects, which may lead to nephroprotective effects.

#### 3.13.5. Other Potential Risk Factors

There is not any clinical literature available to identify risk factors of AKI and tetracycline use.

#### 3.13.6. Summary

There was not a systematic evaluation of tetracyclines that reported an increased risk of AKI. The potential nephroprotective effect of minocycline deserves further investigation.

### 3.14. Fluoroquinolones (FAERS ROR 1.71, 95%CI 1.49–1.97)

#### 3.14.1. Mechanism of Action

Fluoroquinolones work as topoisomerase inhibitors to inhibit the bacterial DNA replication cycle.

#### 3.14.2. Common Uses

Fluoroquinolones have been widely prescribed for many infections including CAP, HAP, IAI and urinary tract infections, due to their ease of administration and broad coverage. 

#### 3.14.3. Clinical Data Regarding AKI

AKI has been documented with fluoroquinolones in a nested cohort study (rate ratio 2.19, 95% CI 1.74–2.73) [144]. However, the absolute increase in AKI incidence was 6.5 events per 10,000 person-years of fluoroquinolones use. A self-controlled case series also found that there was no significant association [145]. 

#### 3.14.4. Proposed Mechanism of AKI

Mechanisms of AKI have been discussed for ciprofloxacin and levofloxacin. One potential mechanism is the development of crystal nephropathy within either the tubules or interstitial glomerular tissue within the nephron. Ciprofloxacin is also thought to induce granulomatous interstitial nephritis [146,147]. 

#### 3.14.5. Other Potential Risk Factors

The existing literature has not focused on identifying other risk factors in AKI occurring with fluoroquinolone use. Older age (over 65 years) and oliguria are common themes in case reports with AKI and fluoroquinolones [147,148,149]. 

#### 3.14.6. Summary

Fluoroquinolones may result in a small increased risk of AKI. The numerous safety warnings for other fluoroquinolone associated adverse events far outweigh AKI as reasons to potentially not prescribe fluoroquinolones.

### 3.15. Combination Antibiotics

Most of the data regarding combination antibiotic therapy and AKI involves vancomycin. Most investigators have found the concomitant use of piperacillin–tazobactam to be associated with an increased risk of AKI compared to vancomycin monotherapy. There is debate regarding whether this definitional increase in AKI incidence is real or if it is actually an unharmful increase due to increased tubular secretion caused by piperacillin–tazobactam.

Cefepime and meropenem are the other most studied beta-lactams with vancomycin. Both of these agents have been shown to have lower AKI rates than piperacillin–tazobactam when each is used in combination with vancomycin. In general, both cefepime and meropenem have had similar rates of AKI in these combination studies.

There are several studies that have evaluated the use of various antimicrobials in combination with polymyxins for resistant Gram-negative pathogens. However, the majority of these have not used a systematic definition of AKI and have been primarily concerned with whether the combination was more effective than polymyxin monotherapy. Outside of aminoglycoside combinations, which would be expected to increase AKI risk, we are unaware of any studies showing an increased AKI risk with another agent being combined with a polymyxin compared to polymyxin monotherapy. There is the potential that minocycline may be nephroprotective, as discussed earlier.

## 4. Conclusions

Numerous antibiotics contribute to the development of AKI. Not only do individual agents have direct effects on the kidney, but patients receiving antibiotics are also often at inherently high risk for AKI due to underlying condition and illness acuity. Clinicians need to be cognizant when prescribing antibiotics associated with an increased risk of AKI and attempt to limit the long-term use of those agents to the best of their ability and mitigate modifiable risk factors if clinically able (See Table 4). This review highlights many different antibiotics, their specific mechanisms for contributing to AKI, and associated risk factors to help providers identify patients potentially at risk for developing AKI; however, many questions regarding treatment and mitigation of risk factors still need to be further explored to continue to improve outcomes in our patients requiring antibiotic therapy. 

## 5. Future Directions and Expert Opinions

Potential mechanisms to decrease the risk or severity of AKI include antimicrobial stewardship, biomarkers, and alternative dosing strategies. Other experts commonly suggest preventing AKI by identifying patients at high risks and minimizing exposure to nephrotoxic agents. 

Antimicrobial stewardship programs are mandated by Center for Medicare and Medicaid Services, with guidance from the Centers for Disease Control and Prevention, and have been shown to significantly reduce other adverse effects such as C. difficile infections [150]. One specific example is the use of MRSA nasal swabs to reduce the duration of vancomycin, which has been associated with AKI by numerous studies. Likewise, the lack of a Gram-negative pathogen can result in the discontinuation of the empiric antimicrobial used for that coverage. This de-escalation is currently performed in practice, a main component in stewardship programs currently, which can decrease the length of time on nephrotoxic antibiotic, thereby decreasing AKI risk. This is common with piperacillin–tazobactam, which has been associated with AKI by several studies. Antimicrobial stewardship programs have contributed to decreasing nephrotoxicity rates by assessing the patient’s risk for AKI and minimizing the use of nephrotoxic antibiotics as microbiological cultures are obtained [151]. With increased use of rapid diagnostics, this has allowed antimicrobial stewardship programs to efficiently reduce the length of time on broad-spectrum antibiotics by approximately 3.6 days, which decreased AKI incidence and other adverse events [152].

There has been increasing interest in using biomarkers to assess not only infectious and inflammatory processes, but also their use in AKI prevention and mitigation strategies [153]. There have been thirteen different biomarkers discovered that are associated with AKI. Most are being evaluated exclusively in the critically ill and surgical populations, but are finally being evaluated in patients after years of in vitro and animal studies to determine their potential utility [153]. Procalcitonin levels are starting to be utilized more in clinical practice, especially with treating pneumonia and determining length of antibiotic therapy [154]. Cystatin C is another biomarker that has been specifically studied in different antibiotics and populations as well and has some data in identifying antibiotic-induced AKI [13,155]. In a recent analysis, cystatin c is correlating better with estimated GFR compared to serum creatinine (r = 0.832, *p* < 0.001 vs. r = 0.425, *p* = 0.002) [156]. Neutrophil gelatinase-associated lipocalin, insulin-like growth factor-binding protein 7, and tissue inhibitor of metalloproteinases-2 have been identified in serum and urine as biomarkers with significant AKI prediction abilities in patients who are critically ill, septic, undergoing cardiac surgery, experience trauma, or have diagnosed contrast nephropathy [13,157,158,159,160]. Numerous issues may limit the broader application of these positive findings including the inability to reliably differentiate between pre-renal and intrinsic AKI. Other factors that impact the normal range of these biomarkers and their potential clinical validity include comorbid conditions, chronic inflammation, age, and gender. Another challenge is the increased costs associated with using biomarkers and additional testing to establish individualized and clinically useful results within an institution’s patient-specific population. Time will tell whether additional evidence regarding the impact of these biomarkers on drug dosing and clinical outcomes will determine whether any biomarker will replace serum creatinine as the standard of care [13,157,161].

As outlined above, the majority of alternative dosing strategies to help minimize the risk of AKI have been focused on vancomycin and aminoglycosides. Recent data have clearly shown that AUC-based dosing of vancomycin results in lower AKI rates than high trough (15–20 mcg/mL) dosing. Other experts suggest that neither approach is needed, as the superior efficacy of either goal has not been evaluated in a randomized, controlled trial. Extended-interval aminoglycoside dosing has clearly reduced AKI rates and is the standard of care for any eligible patient who requires an aminoglycoside for a Gram-negative infection. Additional pharmacokinetic and pharmacodynamic studies are needed to determine potential exposure thresholds for precision dosing to minimize AKI risks with the other antibiotics associated with AKI. We have seen the beginnings of this approach with neurotoxicity and beta-lactams, specifically cefepime [162].

A particularly interesting future direction is to improve the prediction of AKI before it occurs. There are teams of scientists developing tools and calculators to better predict risk of AKI. Rank and colleagues are testing a recurrent neural network (RNN) product which is currently being utilized to monitor AKI risk [162]. Hospitals would have the RNN built into the electronic health record to provide real-time monitoring and act as an alerting system to identify patients with an increasing AKI risk. Clinicians would then be able to provide optimal care early to minimize risk and long-term adverse effects of AKI. This approach is also being evaluated by other investigators as well, Tomasev and Mohamadlou, as the use of prediction models can help streamline clinical decision making at the bedside [163,164]. Other experts have suggested that even though these models are simplistic in their development at this time, the progression to using these calculators to develop more accurate tools, and increase utility of biomarkers, can help to create a biomarker panel to use as an initial diagnostic tool to rapidly determine individualized risk factors [165]. This type of model is encouraging as it makes use of the “big data” now available to help improve healthcare in a continuous fashion and the risk scoring can be altered as new risk factors are identified.

In the next few years, vancomycin use may decrease due to (1) the economic challenge of therapeutic drug monitoring and challenges to its impact on effectiveness, (2) other newer anti-MRSA agents being generically available (e.g., linezolid, daptomycin) and/or (3) the potential ability to treat osteomyelitis and endocarditis with one to three doses of a lipoglycopeptide. The appropriate use of any anti-MRSA agent will also likely decline dramatically. These agents will likely be needed in only a small proportion of patients due to MRSA swab testing and other rapid diagnostic approaches.

Over the next five years, we hope that some of the prediction models will have prospective data to determine their potential utility. We believe these models will need to be validated in multiple patient populations and settings to increase generalizability. We also hope that the increasing clinical utilization of cystatin will provide more data about its generalizability and utility in drug dosing and AKI detection. Other biomarkers will likely start being evaluated in patients, but will still likely lag behind cystatin in implementation. Future research is also needed within the clinical-translational field. This research needs to focus on epidemiological studies to confirm the mechanisms by which antibiotics induce AKI and the overall, population-based risk to patients. This information is vital to developing preventative efforts in the future. Finally, the importance of antimicrobial stewardship will only be emboldened by support from Centers for Medicare and Medicaid Services and Infectious Disease Society of America.

## Figures and Tables

**Figure 1 antibiotics-11-01367-f001:**
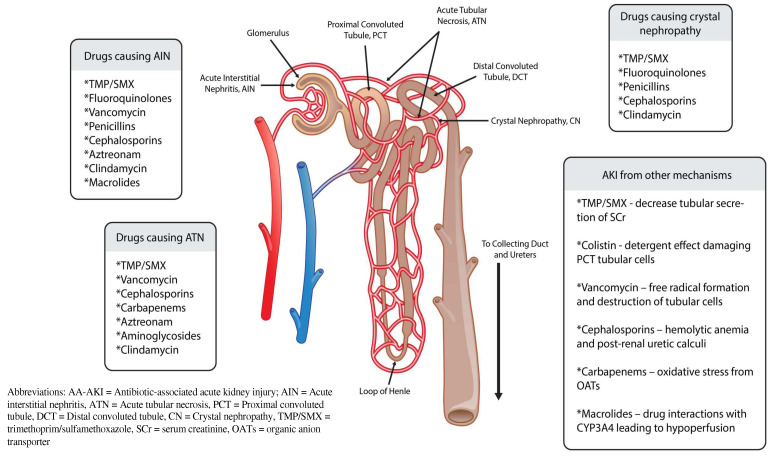
Antibiotics causing AA-AKI and mechanism with nephron.

**Table 1 antibiotics-11-01367-t001:** Classifications of acute kidney injury (per guideline recommendations) [7,8,13,14]. Table adapted with permission from Gameiro, J.; Agapito Fonseca, J., Jorge, S. et al., 2018 [13].

Class/Stage	SCr/GFR			UO		
	RIFLE	AKIN	KDIGO	RIFLE	AKIN	KDIGO
Risk/1 *	↑ SCr × 1.5 or ↓ GFR > 25%	↑ SCr ≥ 26.5 μmol/L (≥0.3 mg/dL) or ↑ SCr ≥ 150 to 200% (1.5–2X)	↑ SCr ≥ 26.5 μmol/L (≥0.3 mg/dL) or ↑ SCr ≥ 150 to 200% (1.5–2X)	<0.5 mL/kg/h (>6 h)	<0.5 mL/kg/h (>6 h)	<0.5 mL/kg/h (>6 h)
Injury/2 *	↑ SCr × 2 or ↓ GFR > 50%	↑ SCr > 200–300% (>2–3X)	↑ SCr > 200–300% (>2–3X)	<0.5 mL/kg/h (>12 h)	<0.5 mL/kg/h (>12 h)	<0.5 mL/kg/h (>12 h)
Failure/3 *	↑ SCr × 3 or ↓ GFR > 75% or if baseline SCr ≥ 353.6 μmol/L (≥ 4 mg/dL) ↑ SCr ≥ 44.2 μmol/L (≥0.5 mg/dL)	↑ SCr > 300% or SCr to 353.6 μmol/L (>4 mg/dL) or initiation of renal replacement therapy	↑ SCr > 300% or SCr to 353.6 μmol/L (>4 mg/dL) or initiation of renal replacement therapy	<0.3 mL/kg/h (>24 h) or anuria (>12 h)	<0.3 mL/kg/h (24 h) or anuria (12 h)	<0.3 mL/kg/h (24 h) or anuria (12 h) or GFR < 35 mL/min/1.73 m^2^ in patients younger than 18 years

Abbreviations: SCr: serum creatinine; GFR: glomerular filtration rate; UO: urine output; RIFLE: Risk, Injury, Failure, Loss of kidney function (dialysis dependence for at least 4 weeks), End-stage kidney disease (dialysis dependence for at least 3 months); AKIN: Acute Kidney Injury Network; KDIGO: Kidney Disease Improving Global Outcomes; mmol/L: micromoles/liter; mg/dL:milligrams/deciliter; mL/kg/h: milliliters/kilogram/hour; h: hours. * indicates: Risk class (RIFLE) corresponds to stage 1 (AKIN and KDIGO), Injury class (RIFLE) corresponds to stage 2 (AKIN and KIDGO), and Failure class (RIFLE) corresponds to stage 3 (AKIN and KDIGO), ↑ increase, ↓ decrease.

**Table 2 antibiotics-11-01367-t002:** Common risk factors for acute kidney injury (AKI) and antibiotic-associated acute kidney injury (AA-AKI).

Risk Factors Associated with Acute Kidney Injury (AKI) and Antibiotic-Associated Acute Kidney Injury (AA-AKI)	
*Hypovolemia* Dehydration Excessive diarrhea or urination (fluid losses)	*Underlying Renal Disease* Elevated baseline SCr Oliguria
*Diuresis* Medically induced (furosemide, torsemide, bumetanide) Non-medically induced	*Critically Ill* Lack of hemodynamic stability Hypotension Lack of renal perfusion
*Exacerbation of comorbid conditions* Congestive Heart Failure Chronic Kidney Disease Diabetes mellitus	*Concomitant Medications and Administration* Lack of dose adjusting for renal dysfunction Drug–Drug interactions Multiple nephrotoxic agents Increased treatment durations with empiric therapies
*Concomitant use of nephrotoxic agents* Contrast dyes ACE-I (lisinopril, enalapril) Angiotensin Receptor-2 Blocker (ARB) (valsartan, losartan) NSAIDs (ibuprofen, naproxen, indomethacin) Antibiotics (aminoglycosides, vancomycin)	*Patient-specific Factors* Older age (>65 years) Increased weight (>91 kg) Recent surgery

Abbreviations: SCr = serum creatinine; ACE-I = angiotensin converting enzyme inhibitor; NSAIDs: non-steroidal anti-inflammatory drug.

**Table 3 antibiotics-11-01367-t003:** Summary of clinical data for antibiotics with highest AKI signals in FAERS.

FAERS Rank	Antibiotic or Class	FAERS ROR	Key Points Regarding AKI
1	Polymyxins	33.1	Reported incidence in peer-reviewed literature: 8–60% Alternatives now available that have less AKI Consensus guidelines for dosing available when necessary
2	Aminoglycosides	17.4	Reported incidence in peer-reviewed literature: 5–25% Frequent monitoring and use of extended interval dosing help minimize the risk of AKI
3	Vancomycin	15.3	More than double the risk of AKI in both a meta-analysis (OR 2.45 vs. 0.3) and a randomized, controlled trial (Event rate: 18.2% vs. 8.4%) Higher trough concentrations are associated with increased AKI risk AUC-targeted dosing is associated with decreased AKI risk
4	Trimethoprim/sulfamethoxazole	13.7	Reversibly inhibits tubular secretion of creatinine, increasing creatinine without affecting GFR Slight increase in AKI risk compared to amoxicillin in a large cohort (2 additional AKI events per 1000 patients) Rate of AKI is dose dependent
5	Penicillin combinations (Beta-lactam/beta-lactamase inhibitor combinations)	8.0	Reported incidence in peer-reviewed literature: 1.7–38.5% Prolonged infusion NOT associated with AKI risk No clear difference in AKI risk between agents in this class
6	Clindamycin	6.5	True AKI incidence is unknown Primary report outside of FAERS is 24 AKI cases in China *
7	Cephalosporins	6.1	Primary observations of AKI risk are case reports Considered an alternative to decrease AKI risk compared to polymyxins or aminoglycosides AKI risk not impacted by prolonged infusion
8	Daptomycin	6.1	Reported incidence in peer-reviewed literature: 5–20% Risk of AKI consistently lower than vancomycin
9	Macrolides	3.6	Reported incidence in peer-reviewed literature: <0.5% AIN is possible as biopsy-proven cases have been published
10	Linezolid	3.5	Reported incidence in peer-reviewed literature: 6–10% Risk of AKI consistently lower than vancomycin
11	Carbapenems	3.3	Lower rates of AKI than polymyxins and piperacillin/tazobactam Cilastatin may have a protective effect vs. AKI
12	Metronidazole	2.6	Only one case report published No known mechanism for AKI
13	Tetracyclines	1.7	AIN has been documented in case reports Minocycline may have a nephroprotective effect
14	Fluoroquinolones	1.7	Absolute AKI risk increase: 6.5 events per 10,000 patient years Be aware of other fluoroquinolone safety risks in patients at risk of developing AKI

* No epidemiological data available as the studies included patients that had clindamycin-induced nephrotoxicity; Abbreviations: FAERS: Food and Drug Administration Adverse Event Reported System; ROR = relative odds ratio; AKI = acute kidney injury; OR = odds ratio; AUC = area under the curve; GFR = glomerular filtration rate; AIN = acute interstitial nephritis.

**Table 4 antibiotics-11-01367-t004:** Key Recommendations.

**Key Recommendation #1:** Clinicians should attempt to identify other patient-specific risk factors for AKI prior to prescribing (or recommending) empiric antibiotic therapy.
**Key Recommendation #2:** If a patient is at increased risk for AKI, the clinician should attempt to utilize antibiotic options which have a decreased risk of, or known incidence of contributing to AKI, when clinically possible.
**Key Recommendation #3:** If a patient does experience AKI due to antibiotic therapy, clinicians must discontinue the offending agent (if known), decrease doses of all other medications as indicated, and determine if other therapy adjustments for treating the infectious insult are required.
**Key Recommendation #4:** If possible, a clinician should attempt to correct, or stabilize, all modifiable risk factors prior to initiating therapy with a nephrotoxic antibiotic.
**Key Recommendation #5:** All clinicians should de-escalate antibiotic therapy once causative pathogens have been identified to potentially limit the duration of nephrotoxic antibiotics.

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
