# Peer review of "The Risk and Clinical Implications of Antibiotic-Associated Acute Kidney Injury: A Review of the Clinical Data for Agents with Signals from the Food and Drug Administration’s Adverse Event Reporting System (FAERS) Database"

_antibiotics, 2022, doi:10.3390/antibiotics11101367_

Round 1

Reviewer 1 Report

This is a good work to comprehensively summarize the association between AKI and antibiotics. It would be better if a Table is used to briefly summarize and compare these 14 antibiotics or antibiotic classes. 

Author Response

We appreciate the thoughtful and thorough provided by the reviewers to improve this work. We have outlined below how we have incorporated their feedback into the manuscript. We are thankful for their time and effort to provide quality peer reviews.

Reviewer 1

  1. This is a good work to comprehensively summarize the association between AKI and antibiotics.

Thank you for complimenting our work.

  1. It would be better if a Table is used to briefly summarize and compare these 14 antibiotics or antibiotic classes.

Thank you for the suggestion. We have now added a table to summarize the numerous antibiotic classes covered for the reader.

Reviewer 2 Report

In this review, Clifford et al. concisely revise antibiotic-associated acute kidney injury with regard to the classes or specific drugs identified within the FAERS reporting system. I believe this review is well-constructed, and the language is clear. I have no further comments besides these minor issues:

  • Can the authors provide further information on the FAERS system? In particular, it is stated that “FAERS is where any adverse event from any medication should be reported by anyone involved with an adverse event”. Does this mean that patients may be involved in the reporting?

  • The acronym ROR (which I suppose stands for reporting odds ratio) should be clarified at least once and explained. 

  • Can the authors provide a table with the 14 antibiotic classes and specific antibiotics and the associated RORs?

Author Response

We appreciate the thoughtful and thorough provided by the reviewers to improve this work. We have outlined below how we have incorporated their feedback into the manuscript. We are thankful for their time and effort to provide quality peer reviews.

Reviewer 2

  1. This review concisely reviews antibiotic-associated acute kidney injury with regard to the classes or specific drugs identified within the FAERS reporting system. I believe this review is well-constructed, and the language is clear.

Thank you for complimenting our work.

  1. Can the authors provide further information on the FAERS system? In particular, it is stated that “FAERS is where any adverse event from any medication should be reported by anyone involved with an adverse event”. Does this mean that patients may be involved in the reporting?

It does mean that patients could be involved in the reporting. We agree that not all readers are familiar with FAERS as a pharmacovigilance tool and have added a paragraph to briefly describe the tool including its strengths and limitations.

  1. The acronym ROR (which I suppose stands for reporting odds ratio) should be clarified at least once and explained.

We agree and have now define ROR on its first use in the manuscript.

  1. Can the authors provide a table with the 14 antibiotic classes and specific antibiotics and the associated RORs?

Thank you for the suggestion. We have now added a table to summarize the numerous antibiotic classes covered for the reader.

Reviewer 3 Report

The review article mentions different drugs with a significant ROR for kidney damage, which was previously described in the article by Patek et al.

Due to the literature search being carried out taking into account another research, it is worth talking a little about how the RORs were obtained or their formula.

The general structure of the article is not the most appropriate.

There are many unreferenced sections, for example, mechanism of action and common uses throughout the manuscript.

The terminologies "Confidence Interval and the Odds ratio" are not homologated; it is necessary to be cited according to the journal.

In line 159, the number appears. Is 56 a reference?

The authors are not correctly cited: for example, line 273 "Crelin."

The summary section is not properly used since it mentions some aspects that are not previously mentioned in the text.

In the summary sections 357 to 360, it is mentioned that diarrhea and hypovolemia increase the risk of AKI; however, it does not appear in any of the previous sections.

Line 471-472 "Linezolid is not typically associated with AKI. It is most commonly used in infections 471 that have a baseline risk of AKI" is not mentioned.

Acronyms sometimes appear before they are explained, are not used homologously in the manuscript, or were never defined in the manuscript: PBPs, OAT, ATN, AIN, CAP, COPD, UTIs

Line 461, what is Zephyr?

The section "Future Directions & Expert Opinions" is not homologated with the objective of the study; for example, the focus changes from talking about drugs to talking about biomarkers.

Author Response

We appreciate the thoughtful and thorough provided by the reviewers to improve this work. We have outlined below how we have incorporated their feedback into the manuscript. We are thankful for their time and effort to provide quality peer reviews.

Reviewer 3

  1. The review article mentions different drugs with a significant ROR for kidney damage, which was previously described in the article by Patek et al.

We are unsure of what the reviewer means. We took the RORs directly from the Patek et al article. We have now made this point clearer in the introduction.

  1. Due to the literature search being carried out taking into account another research, it is worth talking a little about how the RORs were obtained or their formula.

It appears that we were not clear in our reporting of the RORs. These RORs are from Patek et al and they are used as a framework to organize the review from highest ROR to lowest ROR. Our review does not try to reanalyze the FAERS data, it is providing a summary of what data have been published in the peer-reviewed literature for these antibiotics or antibiotic classes with high RORs.

  1. The general structure of the article is not the most appropriate.

We understand that this reviewer did not appreciate the current structure of the review. We would appreciate more focused comments if specific changes are desired to provide a more preferred structure.

  1. There are many unreferenced sections, for example, mechanism of action and common uses throughout the manuscript.

We respect the reviewers opinion, but respectfully disagree. We feel this information is common knowledge and it the other reviewer reports appear to agree with this sentiment. That being said, we will add these numerous references if the editor deems that it will add sufficient value to the audience.

  1. The terminologies "Confidence Interval and the Odds ratio" are not homologated; it is necessary to be cited according to the journal.

At this time the authors are unsure of what this reviewer means by “not homologated”. Odds ratios and confidence intervals are numerical results used to report the likelihood of an event occurring and its likelihood to occur again if similar factors occurred.  The authors have provided more explanation with 3.0.1 why this was used to organize the antibiotics in the manuscript.

  1. In line 159, the number appears. Is 56 a reference?

Yes. We have now added this reference. Thank you for catching this error.

  1. The authors are not correctly cited: for example, line 273 "Crelin."

We have double checked this and the author appears to be correct (Crellin) in both the text and the bibliography.

  1. The summary section is not properly used since it mentions some aspects that are not previously mentioned in the text.
    1. In the summary sections 357 to 360, it is mentioned that diarrhea and hypovolemia increase the risk of AKI; however, it does not appear in any of the previous sections.

We respectfully disagree. This is mentioned in Table 2 that is referenced in the section 2 “An Overview of Acute Kidney Injury”

  1. Line 471-472 "Linezolid is not typically associated with AKI. It is most commonly used in infections that have a baseline risk of AKI" is not mentioned.

We respectfully disagree as this is discussed in 3.10.4 and 3.10.5. We did provide a slight wording adjustment in this summary to provide additional clarification.

  1. Acronyms sometimes appear before they are explained, are not used homologously in the manuscript, or were never defined in the manuscript: PBPs, OAT, ATN, AIN, CAP, COPD, UTIs

Thank you for catching this. We have double-checked the manuscript and have now spelled out all abbreviations on their first use.

  1. Line 461, what is Zephyr?

Reference 68 was commonly referred to as the “ZEPHyR trial” in the Infectious Diseases community when it was first published (ie, https://pubmed.ncbi.nlm.nih.gov/22947593/). To other audiences, this name may not have any context and therefore no meaning. We have restructured the sentence to remove the mention of the trials name to help avoid any confusion.

  1. The section "Future Directions & Expert Opinions" is not homologated with the objective of the study; for example, the focus changes from talking about drugs to talking about biomarkers.

Thank you for the comment. We understand this reviewer’s concern about the future directions not explicitly stating specific medication recommendations. A majority of the new literature, that is being performed to identify and prevent AKI, has been looking at biomarkers and creating more “real-time” lab tests to be used in the clinical setting to determine likelihood of AKI occurring and avoiding medications that are known to be nephrotoxic.

Reviewer 4 Report

Well written manuscript, informative and scientifically sound. Can be accept as it is.

Author Response

We appreciate the thoughtful and thorough provided by the reviewers to improve this work. We have outlined below how we have incorporated their feedback into the manuscript. We are thankful for their time and effort to provide quality peer reviews.

Reviewer 4

  1. Well written manuscript, informative and scientifically sound. Can be accept as it is.

Thank you for complimenting our work.

Reviewer 5 Report

In the era of emerging antibiotic use, there is an increase in overall incidence of all-cause AKI. Additionally, a significant percent of patients, especially hospitalized ones, having a non-AA-AKI receive several different types of antibiotic treatment. This review summarizes all mostly used antibiotic groups, their mechanism of action, and association with AKI pathophysiology. On a comprehensive way, it emphasizes most important facts about the role of each antibiotic group in contributing to development of AKI.

This review fits very well into the existing literature filling many gaps with the facts which are often overlooked. However, although relevant, this topic requires further analysis specific for different patients groups which could be a task for future studies. Since there are couple of references focusing on pediatric patients, the authors should consider adding a small section describing the differences in developing AA-AKI with some specific antibiotic groups between adults and children. Selection of references is also acceptable, although due to the broadness of the topic, it requires the analysis of some older findings as well.

Conclusions are coherent with the review text and the section describing current trends in developing new biomarkers of early recognition and prognosis of developing AKI is especially valued. However, adding a table summarizing most important points for each antibiotic group related to the contribution to AKI development should also be considered by the authors.

Author Response

We appreciate the thoughtful and thorough provided by the reviewers to improve this work. We have outlined below how we have incorporated their feedback into the manuscript. We are thankful for their time and effort to provide quality peer reviews.

Reviewer 5

  1. This review summarizes all mostly used antibiotic groups, their mechanism of action, and association with AKI pathophysiology. On a comprehensive way, it emphasizes most important facts about the role of each antibiotic group in contributing to development of AKI. This review fits very well into the existing literature filling many gaps with the facts which are often overlooked.

Thank you for complimenting our work.

  1. Since there are couple of references focusing on pediatric patients, the authors should consider adding a small section describing the differences in developing AA-AKI with some specific antibiotic groups between adults and children.

Thank you for bringing this to the author’s intention. While there are reports of AA-AKI in the pediatric population, the authors intended to focus on only adult populations. At this time, the authors have removed any references to pediatric patients and edited the manuscript. Assessing the risk of AA-AKI in the pediatric population would be an interesting review to pursue in the future.

  1. Conclusions are coherent with the review text and the section describing current trends in developing new biomarkers of early recognition and prognosis of developing AKI is especially valued.

Thank you for complimenting our work.

  1. Adding a table summarizing most important points for each antibiotic group related to the contribution to AKI development should also be considered by the authors.

Thank you for the suggestion. We have now added a table to summarize the numerous antibiotic classes covered for the reader.

Round 2

Reviewer 3 Report

NO MORE COMMENTS